# Factors Affecting Masticatory Satisfaction in Patients with Removable Partial Dentures

**DOI:** 10.3390/ijerph18126620

**Published:** 2021-06-20

**Authors:** Tasuku Yoshimoto, Yoko Hasegawa, Simonne Salazar, Satsuki Kikuchi, Kazuhiro Hori, Takahiro Ono

**Affiliations:** 1Division of Comprehensive Prosthodontics, Faculty of Dentistry, Graduate School of Medical and Dental Sciences, Niigata University, Niigata 951-8514, Japan; yoshimoto@dent.niigata-u.ac.jp (T.Y.); cem17150@dent.niigata-u.ac.jp (Y.H.); kikuchi@dent.niigata-u.ac.jp (S.K.); hori@dent.niigata-u.ac.jp (K.H.); 2Department of Dentistry, Centro Escolar University, Manila 1000, Philippines; simonnesalazar@yahoo.com

**Keywords:** dentures, removable partial denture, mastication, satisfaction, masticatory performance, quality of life

## Abstract

(1) Background: The degree of satisfaction with dental treatment varies among patients, and the discrepancy may exist between the patient’s subjective evaluation and the objective assessment. Further, the optimal methods for increasing patient satisfaction with mastication remain unclear. This study aimed to identify factors affecting masticatory satisfaction in patients with removable partial dentures. (2) Methods: A total of 132 participants (71.0 ± 9.0 years) were included. Masticatory satisfaction was assessed on a visual analog scale. An oral health survey was conducted to assess the number of functional teeth, missing tooth classification: Kennedy classification, occlusal support: Eichner classification, and removable partial dentures wearing jaw. Objective masticatory performance was assessed using gummy jelly, while subjective masticatory ability was assessed using food acceptance status and oral health-related quality of life. The associations of these factors with masticatory satisfaction were assessed. (3) Results: Masticatory satisfaction among removable partial denture wearers was not significantly associated with gender, age, denture wearing jaw, Kennedy classification, and occlusal support. The degree of masticatory satisfaction was significantly greater with higher levels of masticatory function: masticatory performance, food acceptance score, and OHIP-14 score. The OHIP-14 score was the only significant explanatory variable for masticatory satisfaction in the multiple regression analysis; the strongest associations were with the “psychological discomfort” and “physical disability” subscales (*p* = 0.02 and *p* = 0.005, respectively). (4) Conclusions: Masticatory satisfaction among removable partial denture wearers was strongly associated with oral health-related quality of life, in which the ability to eat meals comfortably with removable partial dentures is the most important determinant of masticatory satisfaction.

## 1. Introduction

Masticatory function among patients with removable prostheses has been assessed and reported using a variety of subjective and objective methods [1,2,3] by which functional impairments prior to prosthesis fabrication as well as treatment outcome were determined [4,5]. The assessment of masticatory performance can be divided into subjective and objective methods. In terms of the objective assessment of masticatory performance, masticatory samples, such as the sieving method using peanuts [6], gummy jelly [7], or color-changing chewing gum [8], are generally used to determine masticatory performance scores, which reportedly depend on the number of remaining teeth [9], occlusal support [10], occlusal force [11], number of occlusal contact points [12], occlusal contact area [13], cusp inclination [14], occlusal configuration [15], cognitive state [16], and tongue pressure [17]. Subjective assessments of masticatory ability typically utilize questionnaires to assess which foods can be masticated [18,19,20]. In order to determine the success or failure of removable partial denture (RPD) treatment, patients are typically asked about their food acceptance [18,19] or administered a self-reported questionnaire pertaining to the oral-health-related quality of life (OHRQoL) [21] or RPDs [22,23,24].

Meanwhile, the degree of satisfaction with dental treatment varies among patients, and it is not easy for dentists to confirm patients’ satisfaction accurately [25]. In addition, if the process and results of dental treatment do not improve patients’ satisfaction, it is not possible to build a strong and trusting relationship between the dentist and the patient. A trusting relationship with the patient is important in order to provide effective dental treatment [26]. Therefore, it might be necessary to identify the level of satisfaction of patients with their treatment.

For some patients, satisfaction with dentures relates to comfort, masticatory efficiency, esthetics, retention, and the ability of the patient to adjust to dentures [22,27]. Psychological-related aspects are the greatest factor affecting OHRQoL among elderly denture wearers [22,27,28].

In addition, when focusing on the satisfaction of denture wearers in terms of mastication, Morii reported that masticatory satisfaction was significantly related to masticatory performance, occlusal support, and oral dryness during meals [7]. However, it has also been reported that the subjective and objective assessments of masticatory function deviate as the number of remaining teeth in the patient decreases. Thus, there is still room for consideration of the factors that contribute to the degree of masticatory satisfaction, and the optimal methods for increasing patient satisfaction with mastication remain unclear [29,30].

Moreover, there is still room for consideration of the factors that contribute to the degree of masticatory satisfaction, and the optimal methods for increasing patient satisfaction with mastication remain unclear.

Therefore, in this study, we examined the relationship between the degree of satisfaction with mastication and subjective and objective masticatory function among patients wearing RPDs. Furthermore, we aimed to identify factors that enhance the degree of masticatory satisfaction.

## 2. Materials and Methods

### 2.1. Participants

We designed this retrospective cross-sectional observational study. The patients of this study were registered in the database of a progress prospective observational study that was approved by the Human Research Ethics Committee at Niigata University (Approval Number 2015-3038) and registered in the University hospital Medical Information Network (UMIN) Clinical Trials Registry (UMIN000043338). The study was performed in accordance with STROBE guidelines. The patients of this study who were excerpted visited the Removable Prosthodontic Clinic or General Dentistry and Clinical Education Unit at Niigata University Medical & Dental Hospital between May 2015 and July 2018 and routinely wore maxillary and/or mandibular RPDs.

The inclusion criteria were as follows: participants who routinely wore maxillary and/or mandibular RPDs; and those who did not complain of any pain or functional disturbances; no known history of temporomandibular joint disorders; and able to understand and appropriately respond to a questionnaire. Patients were excluded for the following reasons: limited usage of their RPDs (e.g., wearing them only during meals or typically removing them); RPDs used as maxillofacial prostheses; mandibular or maxillary complete dentures; and the presence of third molars. All participants provided written consent after receiving a verbal and written explanation of the study purpose and methods.

### 2.2. Oral Examinations

Assessment of oral health condition was carried out by the dentist affiliated with the Niigata University Medical & Dental Hospital. All dentists were calibrated twice (2 h/day, total of 4 h) before assessment. To avoid measurement bias, dentists that carried out the assessments were blinded to the objective of this study. Dentists affiliated with the Niigata University Medical & Dental Hospital examined the number of functional teeth, duration of RPD use, RPD wearing jaw, missing tooth classification, and occlusal support area in an outpatient dental examination room.

The number of functional teeth was defined as the total number of natural teeth and teeth that were restored with crowns or replaced with bridge pontics and implants; retained roots and third molars were excluded [31]. Participants were classified into three groups according to RPD wearing jaw: the maxilla, mandible, and maxilla and mandible.

Missing tooth classification was assessed per arch in each case, using the Kennedy classification: class I, a bilateral edentulous area situated posterior to natural teeth; class II, a unilateral edentulous area situated posterior to natural teeth; class III, an edentulous space bounded on both sides by natural teeth; and class IV, a single bilateral edentulous area located anterior to natural teeth [32]. Occlusal support was assessed with the Eichner classification [33], which comprised three groups: A-group, with posterior occlusion; B-group, without posterior occlusion; and C-group, without occlusal contact regions.

### 2.3. Assessment of Masticatory Satisfaction

Participants’ satisfaction with mastication was defined as their response on a visual analog scale (VAS) to the following question: “How do you feel about chewing comfort with your removable dentures?” The VAS involved a 100-mm-long line printed on paper, with “0” being “completely dissatisfied with the bite” and “100” being “completely satisfied with the bite.” [34]. Before the VAS examination, the patient was shown an example and told how to answer. The patient was asked to fill out the form in a position close to the state he or she considered. The distance from the right end to the marked cross was evaluated to indicate the degree of subjective satisfaction of the patient with their RPD.

### 2.4. Assessment of Objective Masticatory Performance

Masticatory performance was evaluated using a fully automated measuring device developed by Nokubi et al. [35]. All evaluations were performed by a supervising technician, who also relayed instructions to the participants. The participants were instructed to masticate a piece of test gummy jelly (dimensions: 20 × 20 × 10 mm; weight: 5.5 g; UHA Mikakuto, Osaka, Japan) 30 times, and subsequently, expectorate the masticated fragments onto a piece of gauze spread over a paper cup [35]. Any saliva adhering to the surfaces of the gauze or comminuted gummy jelly pieces was removed by running water over the surfaces for 30 s. Next, the retrieved gummy jelly fragments were inserted into the masticatory performance measurement device (Tokyo Photoelectric Co., Ltd., Tokyo, Japan) [35]. The increase in the surface area of the comminuted gummy jelly piece (mm^2^) was determined, and the result was used as the masticatory performance value.

### 2.5. Assessment of Food Acceptance

The questionnaire developed by Sato et al. [19], was used to determine the circumstances of ingesting 20 different food items (Table 1).

The questionnaire regarding the state of food acceptance includes instructions to categorize the chewing difficulty of 20 different food items, using the following symbols:Foods that are easy to chew: (〇)Foods that are difficult to chew: (Δ)Foods that are impossible to chew: (×)Foods that they do not eat because of dislike/no experience in eating: (−)

(〇) is scored 1 point, (Δ) and (×) are scored 0 points, and foods marked with (−) are subtracted from the denominator [19]. Thus, the following expression is used to calculate the percentage of total Food Acceptance Score (FAS) points for each food [19].

FAS (%) = (number of foods that are easy to chew/[number of the 20 foods written on the FAS questionnaire-number of foods marked as (−)]) × 100.

The 20 food items are classified into five groups based on a masticatory index that indicates the ease or difficulty of mastication. The chewing index is derived from the proportion of patients with complete dentures (*n* = 110) who can eat normal-sized food items, similarly to patients with natural dentition [19]. The categorization of the five groups, as based on this masticatory index, is as follows: group 1 (masticatory index of ≥81), group 2 (masticatory index of 61–80), group 3 (masticatory index of 41–60), group 4 (masticatory index of 21–40), and group 5 (masticatory index of ≤20). Patients in group 5 have the greatest difficulties with mastication, while patients in group 1 can masticate with ease.

The FAS questionnaire asks participants to categorize the mastication of 20 different food items with the following symbols: (〇) for foods that are easy to chew; (△) for foods that are difficult to chew; (×) for foods that are impossible to chew; and (－) for foods that they do not eat because they dislike/have no experience in eating them. Each symbol is then awarded points according to the following scheme: 1 point (〇), and 0 points (Δ and ×). Foods marked with (－) are subtracted from the denominator. Thus, the following expression is used to calculate the percentage of total FAS points for each food:

### 2.6. Assessment of OHRQoL

Participants’ OHRQoL was assessed using the OHIP-14, an abridged version of the OHIP [36,37] (Table 2).

The OHIP-14 is classified into seven subscales: functional limitation, pain, psychological discomfort, physical disability, psychological disability, social disability, and handicap. Each subscale includes two questions, which are answered as relevant within the past one month. Responses range from 0 to 4: very often = 4, fairly often = 3, occasionally = 2, hardly ever = 1, and never = 0. The scores are summed to obtain the total OHIP score (highest possible score = 56 points), with higher scores reflecting lower OHRQoL.

### 2.7. Statistical Analysis

All data were assessed using a normality test (Shapiro-Wilk test) and a test of homogeneity of variance, after which the appropriate statistical method was selected. For data with a non-normal distribution, logarithmic transformation was performed.

The null hypothesis was: “Masticatory satisfaction of patients with a removable partial denture is not related to subjective or objective evaluation assessment of masticatory function.” Participants were classified by gender, RPD wearing jaw, Kennedy classification, and Eichner classification, and each subgroup’s masticatory satisfaction was compared using the Mann-Whitney U-test or Kruskal-Wallis test. Spearman’s correlation coefficient was used to study the relationships between masticatory satisfaction and each of the measured items.

To examine factors that affected masticatory satisfaction, a multiple linear regression analysis was carried out, with masticatory satisfaction as the objective variable. Explanatory variables included masticatory performance, the FAS, and the OHIP-14 score; gender and age were moderating variables. In order to examine the relationship between masticatory satisfaction and the OHIP-14 score, an additional multiple linear regression analysis was carried out with masticatory satisfaction as the objective variable and the seven OHIP-14 subscales as explanatory variables; gender and age were moderating variables. Statistical analysis was performed with SPSS Statistics 24 (IBM, Tokyo, Japan), and the level of statistical significance was set at 5%.

## 3. Results

### 3.1. Participant Characteristics

Table 3 presents the participant characteristics. This study recruited 132 patients (55 men, 77 women; mean age, 71.0 ± 9.0 years).

Approximately 50% of the participants wore both maxillary and mandibular RPDs. In terms of the missing tooth classification, Kennedy classification II: i.e., unilateral missing posterior teeth, was the most common category (over 40%) for both maxillary and mandibular arches. In terms of occlusal support, the Eichner B group: i.e., without some occlusal contacts in the molar region, accounted for >80% of cases.

### 3.2. Masticatory Satisfaction

The mean masticatory satisfaction score, as assessed by a visual analog scale (VAS), was 75.3 across all participants; 70% of participants had a VAS score of ≥70, while 17% of participants had a VAS score of <50 (Figure 1).

There was no difference in masticatory satisfaction between subgroups defined by gender, RPD wearing jaw, Kennedy classification, or Eichner classification (Table 4).

Masticatory satisfaction was found to be positively correlated with masticatory performance, as assessed using gummy jelly and the food acceptance score (FAS), and negatively correlated with the Oral Health Impact Profile (OHIP-14) score (Table 5).

Satisfaction was more strongly correlated with the OHIP-14 score than with masticatory performance and the FAS. All FAS subgroups, with the exception of group 1, exhibited a significant correlation with masticatory satisfaction. All seven subscales of the OHIP-14 exhibited a significant correlation with masticatory satisfaction; however, these correlations were weak, with the exception of that for the physical pain subscale.

### 3.3. Factors Contributing to Masticatory Satisfaction

Based on the results of the multiple linear regression analysis, only the OHIP-14 score was identified as a significant explanatory variable for masticatory satisfaction (Table 6).

Masticatory satisfaction was the objective variable in the multiple linear regression analysis; MP, FAS, and OHIP-14 were the explanatory variables; gender and age were moderating variables.

According to this result, an additional multiple regression analysis was subsequently performed with the OHIP-14 subscales as explanatory variables; scores for the psychological discomfort and physical disability subscales were found to be significantly associated with masticatory satisfaction (Table 7).

## 4. Discussion

Multiple linear regression analysis showed that OHRQoL, particularly in terms of self-perceived psychological disability and physical disability, had a significant impact on masticatory satisfaction. Previous reports have indicated that satisfaction with RPDs is the strongest predictive factor of OHRQoL, which supports our results [38,39]. This finding clarifies the meaning of masticatory satisfaction as evaluated by the VAS and suggests methods for increasing patients’ satisfaction with RPDs.

The VAS method allows quantitative assessment of subjective opinion, and since it is a continuous variable, it has an advantage in that it can also be converted to a categorical variable [40]. Nevertheless, one drawback is that patients cannot respond if they do not fully understand the method of responding. The participants in the present study were elderly individuals who were able to attend an outpatient visit at a university hospital and had no problems with cognitive function. They were therefore deemed able to respond appropriately. VAS was originally used to quantitatively measure patients’ pain, but recently it has also been used to quantitatively measure patients’ subjective opinions. Lamb et al. had patients evaluate the effect of relining a mandibular full-bed denture using the VAS and stated that it was effective in quantitatively measuring their subjective opinion of the denture [41]. It was also reported that there was no difference in results when the VAS was compared with the multiple-choice questions [42]. In this study, evaluation of masticatory satisfaction by VAS was found to be related to the subjective masticatory ability by the OHIP questionnaire. Because of this, the use of the VAS scale may be faster and easier to evaluate subjective masticatory ability than a long-winded questionnaire [43].

The result of this study concurs with that of previous studies, which did not report a strong relationship between subjective and objective evaluations of masticatory function [44,45]. According to Salazar et al., the presence or absence of molar occlusion may also yield different relationships in subjective and objective assessments; thus, an accurate and comprehensive assessment of masticatory function must consider both subjective and objective perspectives [46].

The difference in the strength of the relationship between masticatory satisfaction and both the FAS and OHRQoL may reflect the different content and subjective perspectives that these assessments capture. Food acceptance assessments, such as FAS, assess masticatory function based on the ability to masticate specific food items. Although the “masticate without problems” condition is an assessment based on individual criteria, it is considered a more concrete assessment of masticatory function.

The results of the present study suggested that these factors, particularly the psychological discomfort and physical disability OHRQoL subscales, were more significantly associated with masticatory satisfaction. It is known that OHRQoL is related to patients’ clinical oral health, as well as their social and mental/psychological background [45]. The psychological discomfort OHRQoL subscale indicates that it is necessary to consider “appearance from others and nervousness when wearing dentures”. As meals are often taken in a social setting while talking face-to-face with others, RPDs with both satisfactory esthetics and function are required for high masticatory satisfaction. The physical disability OHRQoL subscale indicates that “dissatisfaction with eating with dentures” needs to be considered as well. According to Gordon et al., the nutritional status is more affected by self-assessment of mastication status than the actual dental status [47].

That is, patients wearing RPDs who feel they have problems with their mastication tend to eat high-calorie diets that are easy to chew and swallow and low protein intake. It is generally reported that wearing a denture improves nutritional intake [48]. However, if there is pain and dissatisfaction with the use of dentures, it is expected that the use of dentures will not lead to improvement in nutritional intake [49]. It has been reported that undernutrition in the elderly causes the onset of infections and the need for long-term care, leading to an increase in mortality [48].

The implication of this study is that masticatory satisfaction in patients who use RPDs was closely associated with OHRQoL, particularly in terms of psychological discomfort and physical disability. That is, if patients feel that their RPDs allow for “eating with peace of mind” and “the ability to dine together with other people”, this is likely to be reflected in their reported level of masticatory satisfaction. This might be implying that dentists must regularly enquire as to whether a patient has experienced difficulty in eating, particularly in the presence of others, and that this is important regardless of the objective quality of the prosthesis or the patient’s objective masticatory performance.

This study had some limitations. The study patients were limited to hospital patients who provided their consent to participate and excluded patients with denture pain and/or dysfunction. There might have been a selective bias. In addition, study patients were extracted from research-in-progress. The VAS method, which was used to assess the degree of satisfaction in the present study, is a continuous assessment scale. The quantitative assessment of subjective opinions can reflect minute differences, yielding data on interval scale levels if required [40,50]. However, continuous scales are more easily affected by biases from confounding factors, such as patients’ personalities [51]; furthermore, the maximum degree of satisfaction varies among patients. In addition, the assessment of OHRQoL is strongly reflective of individual values and involves a complex interrelationship of social factors, economic situations, and psychological status [45,52]. A range of potential confounding factors (e.g., patients’ social factors, psychological status, and financial situation; RPD design; and the proficiency of the attending doctor) were not assessed in the present study; future studies should examine these variables and elucidate their impact on masticatory satisfaction.

## 5. Conclusions

The present study found that masticatory satisfaction was not closely related to patient factors, such as age, gender, occlusion, or the distribution of missing teeth. Masticatory satisfaction was more strongly associated with OHRQoL than with objective masticatory performance and self-reported food acceptance. The psychological discomfort and physical disability components of OHRQoL were the most strongly associated with masticatory satisfaction; this suggests that the ability to eat meals comfortably with RPDs is the most important determinant of masticatory satisfaction.

## Figures and Tables

**Figure 1 ijerph-18-06620-f001:**
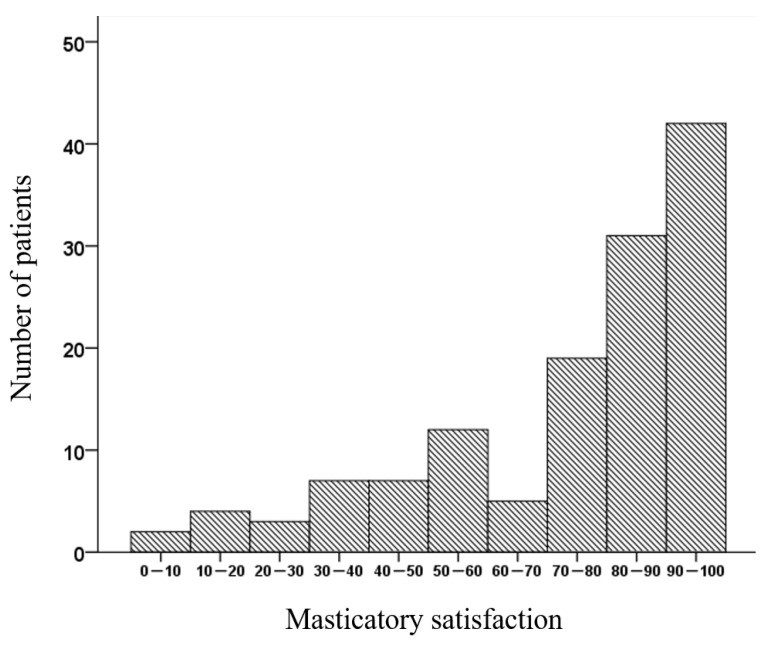
Distribution of masticatory satisfaction. Masticatory satisfaction—The response on a visual analog scale to the question “How do you feel about chewing comfort with your removable dentures?” The visual analog scale ranges from 0 (dissatisfaction with mastication) to 100 (satisfaction with mastication).

**Table 1 ijerph-18-06620-t001:** Food acceptance score questionnaire.

Group	Food	Chewing Index
1	Tofu	93
	Fried egg	89
	Boiled potatoes	88
	Boiled carrots	86
2	Bean sprouts	74
	Fish cakes	74
	Potato chips	66
	Apple	66
	Burdock root	65
3	Arare (bite-sized rice crackers)	60
	Grilled meat	60
	Peanuts	55
	Pickled radish	41
	Hard biscuit	41
4	Hard Japanese crackers	35
	Aged pickled radish	28
	Egg cockle	32
5	Dried squid	17
	Dried scallop	16
	Gum	15

Chewing index: The proportion among 110 complete denture wearers who are able to eat a normal-sized food product in a manner similar to those with natural dentition.

**Table 2 ijerph-18-06620-t002:** Short-form oral health impact profile questionnaire.

How Often Have You Experienced the Problem during the Last Month?	
Functional limitation	1. Have you had trouble pronouncing any words because of problems with your teeth, mouth, or dentures?
	2. Have you felt that your sense of taste has worsened because of problems with your teeth, mouth, or dentures?
Physical pain	3. Have you experienced painful aching in your mouth?
	4. Have you found it uncomfortable to eat any foods because of problems with your teeth, mouth, or dentures?
Psychological discomfort	5. Have you been self-conscious because of your teeth, mouth, or dentures?
	6. Have you felt tense because of problems with your teeth, mouth, or dentures?
Physical disability	7. Has your diet been unsatisfactory because of problems with your teeth, mouth, or dentures?
	8. Have you had to interrupt meals because of problems with your teeth, mouth, or dentures?
Psychological disability	9. Have you found it difficult to relax because of problems with your teeth, mouth, or dentures?
	10. Have you been a bit embarrassed because of problems with your teeth, mouth, or dentures
Social disability	11. Have you been a bit irritable with other people because of problems with your teeth, mouth, or dentures?
	12. Have you had difficulty doing your usual jobs because of problems with your teeth, mouth, or dentures?
Handicap	13. Have you felt that life, in general, was less satisfying because of problems with your teeth, mouth, or dentures?
	14. Have you been totally unable to function because of problems with your teeth, mouth, or dentures?

The questionnaire is classified into seven subscales, with two questions in each subscale. Questions are answered according to the patients’ experience within the past 1 month, using scores of 0 to 4: very often = 4, fairly often = 3, occasionally = 2, hardly ever = 1, and never = 0. The oral health impact profile score (0–56) is the value obtained by summing the score for each question item.

**Table 3 ijerph-18-06620-t003:** Participant characteristic.

Total (*n*)	132
Gender	
Men (%)	55 (41.7)
Women (%)	77 (58.3)
Age (years)	71.2 ± 9.0
Masticatory satisfaction1 (0–100)	75.3 ± 24.2
Functional teeth (*n*)	17.8 ± 6.4
Masticatory performance (mm^2^)	3552.5 ± 1496.0
FAS (0–100)	75.6 ± 23.5
OHIP-14 (0–56)	9.8 ± 8.0
Duration of wearing RPDs (days)	1066.3 ± 1426.0
RPD wearing jaw	
Maxilla	35 (26.5)
Mandible	32 (24.2)
Maxilla and mandible	65 (49.3)
Kennedy classification	
Maxilla	
I	26 (26.0)
II	56 (56.0)
III/IV	18 (18.0)
Mandible	
I	33 (34.0)
II	53 (54.6)
III/IV	11 (11.4)
Eichner classification	
A	6 (4.6)
B	108 (81.8)
B1/B2/B3/B4	31/33/19/25
C	18 (13.6)

Data are presented as the mean ± standard deviation or number of patients. Masticatory satisfaction—The response on a visual analog scale to the question “How do you feel about chewing comfort with your removable dentures?” Functional teeth—Natural teeth and treated teeth that have a crown, as well as bridge pontics and implants. Masticatory performance—The increase in the surface area of the masticated gummy jelly piece (mm^2^). FAS—Food Acceptance Score; the questionnaire regarding the state of food acceptance. OHIP-14—Oral Health Impact Profile; assessment of oral-health-related quality of life. RPD—Removable partial denture. Kennedy classification—Missing tooth classification, which comprises four groups per arch. Eicher classification—Occlusal support classification, which included three groups.

**Table 4 ijerph-18-06620-t004:** Relationship between masticatory satisfaction and survey items.

	Masticatory Satisfaction	*p*
Gender		
Men (*n* = 55)	76.0 ± 24.4	0.38
Women (*n* = 77)	74.2 ± 24.0	
RPD wearing jaw		
Maxilla (*n* = 26)	73.7 ± 29.1	0.78
Mandible (*n* = 56)	75.8 ± 25.7	
Maxilla and mandible (*n* = 18)	75.8 ± 20.6	
Kennedy classification		
Maxilla		
I (*n* = 26)	76.5 ± 24.0	0.71
II (*n* = 56)	75.8 ± 81.0	
III/IV (*n* = 18)	70.9 ± 24.8	
Mandible		
I (*n* = 33)	77.9 ± 23.3	0.69
II (*n* = 53)	74.0 ± 22.9	
III/IV (*n* = 11)	78.3 ± 16.6	
Eichner classification		
A (*n* = 6)	83.7 ± 33.9	0.33
B (*n* = 108)	75.9 ± 23.7	
C (*n* = 18)	68.5 ± 23.7	

Data are presented as the mean ± standard deviation or number of patients. Kennedy classification—Missing tooth classification, which comprises four groups per arch. Eicher classification—Occlusal support classification, which included three groups. RPD—removable partial denture. *p*-value—Comparison of masticatory satisfaction in each subgroup by the Mann-Whitney U-test or Kruskal-Wallis test.

**Table 5 ijerph-18-06620-t005:** The correlation between masticatory satisfaction and survey items.

	Masticatory Satisfaction	
	Correlation Coefficient	*p*
Age	0.02	0.80
Number of functional teeth	0.07	0.46
Duration of wearing RPDs	−0.12	0.19
MP	0.20	0.02
FAS	0.41	<0.001
Group 1	0.10	0.24
Group 2	0.38	<0.001
Group 3	0.39	<0.001
Group 4	0.37	<0.001
Group 5	0.28	<0.001
OHIP-14	−0.50	<0.001
Functional Limitation	−0.35	<0.001
Physical pain	−0.51	<0.001
Psychological discomfort	−0.38	<0.001
Physical disability	−0.48	<0.001
Psychological disability	−0.33	<0.001
Social disability	−0.33	<0.001
Handicap	−0.38	<0.001

Functional teeth—Natural teeth and treated teeth that have a crown, as well as bridge pontics and implants. RPD—removable partial denture. MP—masticatory performance; the increase in the surface area of the masticated gummy jelly piece (mm^2^). FAS—Food Acceptance Score; the questionnaire regarding the state of food acceptance. OHIP-14—Oral health impact profile; assessment of oral-health-related quality of life. *p*-values—Spearman’s rank correlation test.

**Table 6 ijerph-18-06620-t006:** Result of the multiple linear regression analysis.

	Partial Regression Coefficient	Standardization Coefficient	Standard Error	95% CI			*p*
Gender	1.2	0.06	3.8	−6.3	to	8.6	0.76
Age	4.3	−0.06	7.3	−10.1	to	18.7	0.56
MP	0.9	−0.08	4.8	−8.4	to	10.3	0.85
FAS	−3.8	0.05	3.3	−10.2	to	2.7	0.25
OHIP score ^a^	−20.8	0.50	5.2	−31.0	to	−10.6	<0.001

^a^—Statistically significant association. MP—masticatory performance; the increase in the surface area of the masticated gummy jelly piece (mm^2^). FAS—Food Acceptance Score; the questionnaire regarding the state of food acceptance. OHIP score—Oral health impact profile; assessment of oral-health-related quality of life. *p*-values—A multiple linear regression analysis was used. Gender was a dummy variable, where 1 indicates male and 0 indicates female. CI—confidence interval.

**Table 7 ijerph-18-06620-t007:** Association between oral health-related quality of life subscales and masticatory satisfaction.

	Partial Regression Coefficient	Standard Error	95% CI			*p*
Gender	0.90	3.9	−6.8	to	8.5	0.83
Age	7.00	7.0	−6.7	to	20.7	0.32
Functional limitation	4.40	8.7	−12.6	to	21.3	0.62
Physical pain	−13.5	9.6	−32.3	to	5.2	0.16
Psychological discomfort ^a^	−27.2	11.7	−50.0	to	−4.3	0.02
Physical disability ^a^	−32.8	11.6	−55.6	to	−10.1	0.005
Psychological disability	15.40	13.5	−11.1	to	41.8	0.25
Social disability	12.60	13.9	−14.7	to	39.8	0.37
Handicap	−13.1	12.6	−37.8	to	11.6	0.30

^a^—Statistically significant association. *p*-value—A multiple linear regression analysis was used. Masticatory satisfaction was the objective variable; the seven subscales of the oral health impact profile-14 were the explanatory variables; gender and age were the moderating variables; gender was a dummy variable, where 1 indicates male and 0 indicates female. CI—confidence interval.

## Data Availability

The materials described in the manuscript, including all relevant raw data, will be freely available to any scientist wishing to use them for noncommercial purposes, by contacting the corresponding author, without breaching patient confidentiality.

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
