# Peer review of "Factors Affecting Masticatory Satisfaction in Patients with Removable Partial Dentures"

_ijerph, 2021, doi:10.3390/ijerph18126620_

Round 1

Reviewer 1 Report

The Authors must see my remarks

The article needs MODERATE revision

Author Response

The Authors must see my remarks
The article needs MODERATE revision
Response: Thank you for your helpful peer review. Accordingly, we have written the answers to each of your points in the notes of the attached PDF. We have also attached the word file that contains the answers to each question. Please check the word file for corrections to the main text. 

Reviewer 2 Report

The introduction is well written but overreports the experience of the authors with RPD patients. I believe that there is some information that should be supported by references, namely in paragraphs 2 and 4.

The materials and methods section is well written and detailed, allowing, as desired, its reproduction. The statistical methods applied are adequate.

The discussion is very well written.

There is no need to repeat the purpose of the study at the conclusion.

Author Response

The introduction is well written but overreports the experience of the authors with RPD patients. I believe that there is some information that should be supported by references, namely in paragraphs 2 and 4.

Response: Thank you for your helpful peer review. According to your suggestion, we have revised the main text to be factual rather than experiential (page 2, lines 8-13, lines 15-22). We have also added the necessary references for each revised text.

There is no need to repeat the purpose of the study at the conclusion.

Response: Thank you for your helpful peer review. According to your comment, the text has been revised (page 10, line 46).

Reviewer 3 Report

Thank you for the opportunity to review this manuscript. In general terms, this study makes an important contribution to the scientific literature. 

I recommend to the authors to add the STROBE guidelines in order to guarantee the follow the quality of the report of the manuscript.

In the study design, please be more specific in the study. For me, it is unclear why this study was approved as a clinical trial study (is it an ongoing project?)

Similarly, I would like to invite the authors to review the limitations of the study related to the type and size of the sample. 

Author Response

I recommend to the authors to add the STROBE guidelines in order to guarantee the follow the quality of the report of the manuscript.

Response: Thank you for your helpful peer review. As you suggested, we have added the mentions in the Material and Methods (page 2, lines 40-41).

In the study design, please be more specific in the study. For me, it is unclear why this study was approved as a clinical trial study (is it an ongoing project?)

Response: As suggested, we have revised the text with more details about the research design of this study. This study is a retrospective analysis and summary of the results of a study related to masticatory satisfaction among the collected data between May 2015 and July 2018. The original prospective observational study was registered in UMIN CTR (UMIN000043338). These details are written in the text (page 2, lines 36-39).

I would like to invite the authors to review the limitations of the study related to the type and size of the sample.

Response: As suggested, we have added limitations of sample size in the “Limitation” section (page 10, lines 30-32).

Round 2

Reviewer 1 Report

Some corrections are required

(The Authors must see my remarks)

Author Response

Reviewer 1

Some corrections are required

(The Authors must see my remarks)

Response: Thank you for your helpful peer review. However, the suggestion you have recently mentioned was already revised in the previous manuscript. We checked the new suggestions and written the answers to clarify each of your points. Please see the attached file. The revised text is indicated in red font in the manuscript for your reference.

[xx1] Reference(s)?

[xx 2R1] The inclusion and exclusion criteria can be referenced from the information registered in the UMIN-CTR(https://upload.umin.ac.jp/cgi-open-bin/ctr/ctr.cgi?function=brows&action=brows&recptno=R000049476&type=summary&language=J)

[xx3] Inclusion criteria?

[xx 4R3] This suggestion was previously mentioned in the last review and was already revised. We have added the inclusion criteria on page 2, lines 45-48 for your reference.

[xx5] Reference(s)?

[xx6R5]  This suggestion was previously mentioned in the last review and was already revised. We have added a reference to the Food acceptance score questionnaire as you have suggested.

[xx7] How did the Authors determine the study sample ?

Protocol?

REference(s)?

Selection/REcall biases? Intra-examiner variability(K-Index)?

[xx8R9]  This suggestion was previously mentioned in the last review and was already revised.  We have revised the text with more details about the research design of this study on page 2, lines 36-41 for your reference. To clarify, this study is a retrospective analysis and summary of the results of a study related to masticatory satisfaction among the collected data from May 2015 to July 2018. The original prospective observational study was registered in UMIN CTR (UMIN000043338).

Due to the fact that the study participants were limited to hospital patients who provided their consent to participate and excluded patients with denture pain and/or dysfunction, there might have been a selective bias. In addition, study patients were extracted from research-in-progress. This limitation of study was added to the "Limitations".

As for the intra-examiner variability, the assessment of oral health condition was carried out by the dentist in Niigata University Medical & Dental Hospital. All dentists were calibrated twice (2 h/day, total of 4 h) before the assessment. To avoid measurement bias, the dentists that carried out the assessments were blinded to the objective of this study. Therefore, the Intra-examiner variability  is deemed reliable.

[xx10] Reference(s)

[xx11R12] We have removed the text without evidence, that is, a corresponding reference.

[xx13]  Do not repeat Results, Personal opinions or Conclusions

[xx14R15]  We believe that this statement has important clinical implications for this study and therefore needs to be repeated. We have revised the personal opinion.
